# Effect of Walkability on Urban Sustainability in the Osaka Metropolitan Fringe Area

**Haruka Kato**

Department of Housing and Environmental Design, Graduate School of Human Life Science,
Osaka City University, Osaka 5588585, Japan; haruka-kato@osaka-cu.ac.jp; Tel.: +81-6-6605-2823

**Abstract:** This study aimed to clarify the effect of walkability on urban sustainability, according to the types of residential clusters in the Osaka Metropolitan fringe area. For this purpose, this study analyzed the statistical causal relationship between the Walkability Index and the Ecological Footprint to Biocapacity (EF/BC) ratio of each residential cluster. The EF/BC ratio is the ratio of the ecological footprint of the biocapacity of the residential clusters. As a result, the effect of walkability on urban sustainability was clarified depending upon the types of residential clusters in the Osaka Metropolitan fringe area. Specifically, it was found that the Walkability Index negatively affects the EF/BC ratio in the sprawl cluster. This suggests that, in the sprawl cluster, active efforts to improve the Walkability Index can contribute to the realization of SDGs (Sustainable Development Goals of the 2030 Agenda). However, Walkability Index has a strong positive effect on the EF/BC ratio in the old new-town cluster, etc. For the residential clusters, the results of this study suggested that there is a necessity to improve urban sustainability through approaches other than improving Walkability.

**Keywords:** walkability; ecological footprint; biocapacity; Osaka metropolitan fringe area; sprawl cluster; old new-town cluster

## 1. Introduction

This study aims to clarify the effect of walkability on urban sustainability, according to the types of residential clusters in the Osaka Metropolitan fringe area. Kato, et al. [1] defined walkability as "relevant to residential environments that promote walking or cycling with safety, comfort, and the attractions of daily life." Walkability has been studied as a concept that contributes to the improvement of residents' health, primarily in the field of public health. For example, Cerin, et al. [2] developed a walkability indicator called ANEWS (Abbreviated Neighborhood Environment Walkability Scale). In addition, they discovered that the majority of active individuals who live in areas of high walkability would be healthier than those who live in areas with low walkability. In Japan, Inoue, et al. [3], using the ANEWS indicator, have verified the associations of physical activity with environmental attributes for Japanese adults. Recently, Hino, et al. [4] investigated location data obtained from the Yokohama Walking Point Project. As a result, they clarified that the residents who walk frequently are healthy and that the living environment promotes walking [4].

At present, in the field of urban planning, many researchers have begun design research on living environments with high walkability. For example, Speck [5] proposed 101 characteristics of walkable cities. In Japan, Kato, et al. [1] developed the walkability indicator. Then, Kato, et al. [6] evaluated the "scenario in which vacant land is used," as an alternative of the compact city in the sprawl cluster of the Osaka Metropolitan fringe area. Since 2020, Japanese MLIT (Ministry of Land, Infrastructure, Transport, and Tourism) has called for promoting "walkable city" projects [7]. Through these projects, 260 organizations in Japan have begun to establish walkable cities [8].

This study hypothesizes that walkable cities contribute not only to improving the health of residents, but also to the realization of Goal 11 (Make cities and human settlements inclusive, safe, resilient, and sustainable) in SDGs (Sustainable Development Goals of the 2030 Agenda). In other words, the burden on the environment will be reduced by creating a walkable living neighborhood where the elderly population can live independently in Japan. The SDGs are "the blueprint for achieving a better and more sustainable future for all" [9]. One hundred and ninety-three countries of the United Nations work with 17 goals and 169 targets of the SDGs. Efforts toward the SDGs will not exceed the planetary boundary established by Rockström, et al. [10]. Regarding the planetary boundary, this study focused on land use, which can contribute to the field of urban planning for SDGs. The Ecological Footprint (abbreviated as EF) is the sustainability indicator for land use, which is evaluated for its effectiveness.

The EF indicator was developed by Wackemagel and Rees [11], representing the land area necessary to support a particular lifestyle toward a sustainable recycling society permanently. Moreover, the Global Footprint Network publishes assessment tools and worldwide footprint data [12]. Using the EF, Wackemagel, et al. [13] defined the significance of biophysical assessments for shaping successful economic policies. For the approaches by EF, Galli, et al. [14] communicated the effectiveness of regional-level policies towards the realization of the SDGs in Portuguese cities. For example, Lee and Peng [15] calculated EF in Taiwan from 1994 to 2011 and proposed measures to help Taiwan advance toward sustainability, carbon reduction, and energy-saving policies that will boost their economy. Shi, et al. [16] analyzed the relationship between EF and the human development index (HDI) in Hong Kong and Singapore during the years 1995–2016. As a result, they provided policy suggestions for transforming toward a "high HDI and low footprint" [16].

Based on previous studies focusing on the concept of ecological footprints, the novelty of this study is that it clarifies the causal relationship between walkability and the Ecological Footprint to Biocapacity (abbreviated as EF/BC) ratio, according to residential cluster types. The EF/BC ratio is the ratio of the ecological footprint of the biocapacity of the residential clusters. In addition, the EF/BC ratio represents the environmental load excess ratio. The EF/BC ratio suggests an ecological balance in each residential area. At the metropolitan scale, EF/BC is balanced by urban FP and rural BC. However, since the EF/BC ratio is in excess, we need to lower the urban FP. Speck [17] suggested that improving walkability contributes to urban sustainability. For example, improving walkability contributes to supporting nearby commercial facilities and reducing $CO_2$ emissions from motorized transportation. However, the results were only qualitatively examined, meaning they have not been clarified quantitatively. By quantitatively analyzing the statistical causal relationship between walkability and sustainability, this research will effectively utilize the efforts of companies and local governments, focusing on walkability toward the realization of the SDGs. Furthermore, if results conclude that residential clusters do not have statistical causal relationships, it is expected that there will be the possibility of finding new approaches other than improving walkability.

The statistical causal relationship was analyzed based on a residential neighborhood association (abbreviated as "NA") scale. Japan has hierarchical structures of governance ranging from NA scale to Metropolitan Scale. The reason why this paper analyzed NA scale is that the scale of the foundation based on which residents make decisions about urban planning in Japan.

As a case study, this study selected the Osaka Metropolitan fringe area, which is one of the major metropolitan areas in Asia. The Osaka Metropolitan fringe area is located in the surrounding areas of the Osaka metropolitan area. The Osaka metropolitan fringe area became urbanized rapidly after the 1950s [1]. Especially around the 1960s, the sprawl urbanization was serious because the fields were converted to residential land by a few sheets of land. In order to prevent the sprawl, high-end residential areas were developed as new-towns in the mountains. Over here, there are several types of residential clusters, including the sprawl cluster and the old new-town cluster [1]. Since the residential clusters are facing the problems of population decline and aging, this study analyzed for the Osaka Metropolitan fringe area. This study analyzed Osaka, Kyoto, and Hyogo prefectures in the Osaka Metropolitan area. The three prefectures are city-regions in the metropolitan area, where the population is declining more rapidly than in the Tokyo and Nagoya Metropolitan areas in Japan [18].

Therefore, the Osaka Metropolitan fringe area must develop sustainable cities, even if the population decreases rapidly in the future. In 2025, the Osaka Expo 2025, which aims to build a society working toward achieving SDGs, will be held in the Osaka Metropolitan area. Therefore, the knowledge of this study will be significantly influential, not only in the Osaka metropolitan area but also in Asia.

## 2. Materials and Methods

The method of this study consisted of four steps (Figure 1). First, residential areas were categorized by urban ecological analysis. Second and third, the Walkability Index and Ecological Footprint were categorized. Finally, the statistical causal relationship between the Walkability Index and the Ecological Footprint was analyzed according to the types of residential clusters.

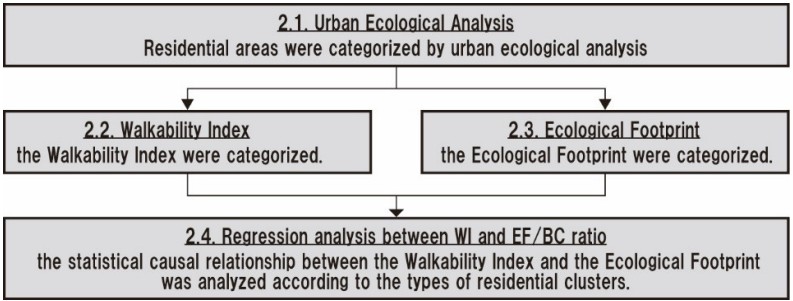

**Figure 1.** Analysis steps in Chapter 2.

### 2.1. Urban Ecological Analysis

The types of residential clusters found in the metropolitan fringe area were clarified by categorizing residential areas using statistical analysis. The categories suggest locations of every type of residential cluster. The method implemented in this section was an urban ecological analysis. The urban ecological analysis is a method of analyzing spatial patterns by an inductive method using a wide range of statistical data [1]. The analysis consisted of five steps.

First, in this section, the standardization of 53 indicators for the NA scale of the Japanese census in 2015 was analyzed [19]. In Japan, the national Census is conducted every five years. Therefore, the 2015 Census is the most recent available data. See Appendix A for percentages of population sorted by age. Next, the standardized composition ratio $R_x^k$ was calculated by standardizing the data of each indicator in the Osaka Metropolitan area in Equation (1). The Osaka Metropolitan area includes Osaka, Kyoto, Hyogo, Nara, Shiga, and Wakayama prefectures. The reason is that the distribution range between the data differs among the 53 indicators.

$$R_x^k = \frac{X_{xi}^k - X_{xmin}}{X_{xmax} - X_{xmin}} \tag{1}$$

where $X_{xi}^k$ is the number of NA *i* for indicator *x* in the residential area *k*.; $X_{xmin}$ is the minimum value of NA *i* for indicator *x*.; and $X_{xmax}$ is the maximum value of NA *i* for indicator *x*.

Third, the principal component was analyzed using the standardized composition ratio $R_x^k$. Since the Cronbach's $\alpha$ coefficient of the principal component analysis was 0.985, reliable data were obtained as social survey data. Fourth, seven principal components were extracted according to the Guttman Kaiser criterion. Reliable data were obtained because the total variable amount of these seven principal components was 78.8%. Finally, residential clusters were categorized by hierarchical cluster analysis using their seven principal component scores. Then, the name of each cluster was determined by analyzing the content composition ratio $R_{ki}$ of each indicator for each residential cluster (Appendix A). Besides, the effectiveness of the clusters was verified from their location on the map and photographs of the actual residential clusters.

Subsequently, to clarify the residential clusters found in the metropolitan fringe area, the "urbanized area ratio" was analyzed. The "urbanized area ratio" is the ratio of the residential areas

included in the urbanized area. Then, "the average distance from the center" of the metropolitan area, which comprises Umeda Station, Karasuma Station, and Sannomiya Station, was calculated. As a result, residential clusters were clarified with over 50% of "urbanized area ratio" and an average city center distance of over 10 km of "the average distance from the center." The analysis clarified the residential clusters located in the Osaka Metropolitan fringe area.

### 2.2. Walkability Index

Residential areas were evaluated using the Walkability Index. Brownson, et al. [20], classified three types of walkability indicators: using GIS-based measures, perceived environment measures, and observational measures. Some of the significant indicators of "Using GIS-based Measures" are Walkability 3Ds and Walk Score®. Walkability 3D was developed as an index to evaluate the impact of the neighboring environment on tourists in the case of the San Francisco Bay coast [21]. Walk Score® was developed as an index to evaluate the walkability of the neighborhood environment in the case of the US metropolitan area [22]. In Japan, Kato, et al. [1] also proposed a Japanese version of a walkability indicator. Among them, the Walkability Index is an evaluation index developed by Frank, et al. [23,24] as an index to evaluate the neighborhood environment in Australia. A feature of this Walkability Index is that many studies have verified its effectiveness on the actual number of pedestrians [25]. In Japan, Kanai, et al. [26] analyzed changes in walking activity due to walking space development using the Walkability Index.

This Walkability Index is composed of the three components, net residential density (abbreviated as "ND") in Equation (2), the density of street connectivity (abbreviated as "SC") in Equation (3), and land use mix (abbreviated as "LUM") in Equation (4). WI, which is the score of the Walkability Index, is the sum of the standardized values of ND, SC, and LUM in Equation (5).

$$\text{ND}_k = \frac{H_k}{A_k} \tag{2}$$

$$\text{SC}_k = \frac{I_k}{L_k} \tag{3}$$

$$\text{LUM}_k = \sum_{i=1}^{3} \frac{p_{k,i} \times \ln p_{k,i}}{\ln n} \tag{4}$$

$$\text{WI}_k = z\text{-ND}_k + z\text{-SC}_k + z\text{-LUM}_k \tag{5}$$

where $H_k$ is the total number of net residents in residential area $k$ by Japanese census data in 2015 [19]; $A_k$ is the total housing area in residential area $k$ by Japanese census data in 2015 [19]; $I_k$ is the number of street connectivity in residential area $k$ by road centerline data [27]; $L_k$ is the total length of the street in residential area $k$ by road centerline data [27] (m); $p_{k,i}$ is the area ratio of land use $i$ in residential area $k$ by the data of the numerical map 5000 in Japan [28]; $i$ is the classification of land use $i$ (residential land, commercial land, public facility land); and z- is the standardized value.

### 2.3. Ecological Footprint

The EF/BC ratio of each residential area was assessed. The EF was calculated by summing up the five footprints (abbreviated as "FP") in Equation (6). Those FP are cropland FP, grazing land FP, forest land FP, urban land FP, and carbon FP. This study calculated EF by referring to the "Ujihara-Taniguchi Model" developed by Ujihara, et al. [29] based on the "compound method." The "compound method" is a top-down method of calculating EF using statistical data of the entire region, and is the opposite methodology to the "component method" that sums up the components of EF. The component method is thought to be problematic because it fails to consider the local variability of EF. However, the compound method produces exceptionally reliable data because the risk of double addition is low [11].

Regarding the "compound method," the "Ujihara-Taniguchi Model" can be analyzed on an NA scale using open data that can be acquired in Japan, and therefore was utilized in this study. In addition, Chen, et al. [30] verified the effectiveness of the model in Japan. The calculations for each FP are provided in Appendix B.

$$EF^k = FP^k_{cropland} + FP^k_{grazing} + FP^k_{forest} + FP^k_{build-up} + FP^k_{Carbon} \tag{6}$$

where $k$ is residential area $k$, $EF^k$ is EF in residential area $k$ (ha), $FP^k_{cropland}$ is cropland FP in residential area $k$ (ha), $FP^k_{grazing}$ is grazing land FP in residential area $k$ (ha), $FP^k_{forest}$ is forest land FP in residential area $k$ (ha), $FP^k_{build-up}$ is build-up land FP in residential area $k$ (ha), and $FP^k_{Carbon}$ is the FP required for $CO_2$ absorption in residential area $k$ (ha).

Using the $EF^k$, this section calculated EF/BC ratio in Equation (7). The EF/BC ratio stands for the environmental load excess ratio. An EF/BC ratio >1.0 suggests that EF is "overshooting" beyond the BC in the residential area $k$. The EF/BC ratio is the best way to analyze the ecologically balanced area, though several methods have been proposed to analyze the relationship between EF and BC. For example, the Global Footprint Network [12] analyzed the difference between EF and BC. However, the difference between EF and BC is not suitable for analyzing the NA scale because it tends to be influenced by the size of the residential areas and its population. Chen, et al. [30] analyzed the local cities of the Tokyo Metropolitan area using the EF/BC ratio. Then, the effectiveness of the index was evaluated because it allows us to analyze the ecological balance of the NA scale without being affected by the size of the residential areas. In 2016, Japan's ratio was 7.7 compared to the global ratio of 1.7. Therefore, Japan has to improve the EF/BC ratio, even in urban areas. For the improvement in Japan, the neighborhood scale of urban central areas needs to lower EF and raise BC, although the Metropolitan scale essentially balances urban EF and rural BC. In Equation (7), $BC^k$ was calculated by Equation (8). $BC^k$ means the bio-productive supply available in the residential area $k$.

$$EF/BC \ ratio = \frac{EF^k}{BC^k} \tag{7}$$

$$BC^k = \sum_{s=1}^{11} bc^k_{lu} \tag{8}$$

where $bc^k_{lu}$ is the area build-up of land use $lu$ in the residential area $k$, which is calculated from the data of the numerical map 5000 in Japan [28]. $lu$ is eleven types of land uses (industrial land, residential land, commercial land, office land, road land, parks, green spaces, public facility land, forest, farmland, and vacant land).

*2.4. Regression Analysis between WI and EF/BC Ratio*

In Section 2.4, the statistical causal relationship between WI and the EF/BC ratio on the NA scale according to the types of residential clusters was analyzed. For this purpose, regression analysis was performed for each residential cluster classified in Section 2.1, by setting WI calculated in Section 2.2 as an explanatory variable and EF/BC ratio calculated in Section 2.3 as an objective variable. Regression analysis was conducted using the stepwise selection method. As a result, this study extracted only the explanatory variables for which significant differences could be confirmed.

## 3. Results

*3.1. Classification of Residential Areas*

Categories of residential clusters were analyzed by the urban ecological analysis. As a result, Appendix A illustrates thirteen types of residential clusters in the Osaka Metropolitan area. These clusters include inter-city cluster, business center cluster, mining industry cluster, dense cluster, public housing cluster, non-residential cluster, agriculture cluster, sprawl cluster, high-rise residential cluster, mountain cluster, old new-town cluster, suburban agriculture cluster, and rural

cluster. Each residential cluster was plotted on the map in Figure 2, which indicates location characteristics according to each residential cluster.

This section clarified the clusters in the metropolitan fringe areas. Thus, residential clusters with over 50% of "urbanized area ratio," and an average city center distance of over 10 km of "the average distance from the center," were identified. As a result, Appendix A illustrates six types of residential clusters in the Osaka Metropolitan fringe areas: dense clusters, public housing clusters, non-residential clusters, sprawl clusters, high-rise residential clusters, and old new-town clusters. Of these six types of residential clusters, the non-residential cluster was excluded from the analysis, because this study analyzed residential areas. Figure 2 verified the same conclusion as a result of urban ecological analysis for the northern part of the Osaka Metropolitan area [1].

Finally, the five types of residential clusters in the Osaka Metropolitan fringe area were identified. First, the dense cluster was defined as densely urbanized residential areas with small wooden houses [31]. The dense cluster was constructed with too narrow streets with fragile road networks, as well as insufficient public open spaces and high population density before the 1950s [31]. The dense clusters have been regarded as dangerous urban areas in Japan, where earthquakes are common [31]. Figure 2 illustrates that the dense cluster is located close to the outer edge of the inner-city cluster. The public housing cluster was defined as residential areas constructed by public institutions for affordable housing. Figure 2 shows that the public housing cluster is dispersed in the metropolitan fringe area. The sprawl cluster was defined as urban residential areas with a lack of urban infrastructure, characterized by narrow streets and small vacant lots, that were developed around farmlands since the 1970s [1]. That sprawl cluster has been regarded as problematic in terms of efficient land use. Figure 2 shows that the sprawl cluster is located between inner-city clusters and old new-town clusters. The high-rise residential cluster was defined as residential clusters constructed as tower-type buildings, constituting redevelopment projects as Transit-Oriented Development. Figure 2 shows the high-rise residential cluster located along the railway and stations. Finally, the old new-town cluster was defined as planned and large-scale suburban residential areas featuring detached houses [32]. The old new-town cluster has encountered urban challenges for older adults, because the cluster was developed at hillside areas with slopes that make it difficult to walk [32]. Figure 2 shows that the old new-town cluster was located on the hillside in the mountains.

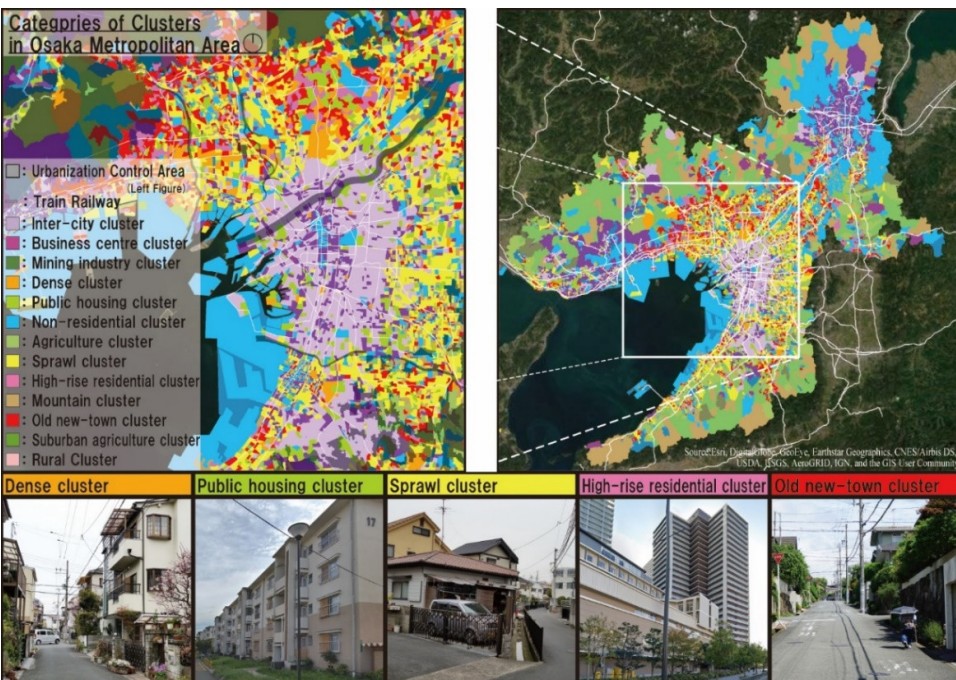

**Figure 2.** Location of the clusters in the Osaka Metropolitan area.

### 3.2. WI and EF/BC Ratio of Each Residential Cluster

The WI and EF/BC ratios of each NA scale were calculated, based on the analyses described in Sections 2.2 and 2.3. Then, the results of WI and EF/BC ratios were illustrated on the map in Figures 2 and 3. Figure 3 categorizes the data by the Jenks natural breaks classification, which is a method of thresholding where the change in the data is relatively large. As a result, Figure 3 shows that the Osaka Metropolitan area has higher WI scores in the suburban areas, compared to Los Angeles Metropolitan area [33]. Besides, Figures 3 and 4 show that the WI and EF/BC ratio of each NA were affected by the residential clusters. For example, Figure 4 shows that the EF/BC ratio is higher in the central areas and lower in the outer edge areas.

Therefore, in Section 3.2, the box plot diagram of the WI and EF/BC ratios according to each residential cluster was analyzed in Figures 5 and 6. As a result, Figures 5 and 6 show that WI has less change in each residential cluster compared to the EF/BC ratio. Regarding the EF/BC ratio, Figure 6 shows that the EF/BC ratio exceeded 1.0 in most clusters. The result means that "overshooting" occurs in the Osaka Metropolitan area. The results show that residents in the Osaka Metropolitan area should alter their lifestyle to lower the EF/BC ratio. Among the clusters, the EF/BC ratio was low in the agricultural cluster (Me = 6.90), and mountain cluster (Me = 4.76). The next highest EF/BC ratios are the residential clusters located in the metropolitan fringe area. These include the sprawl (Me = 12.29) and old new-town clusters (Me = 15.96). Finally, the EF/BC ratio was high in the inner-city (Me = 20.60), dense residential (Me = 20.64), and public housing area clusters (Me = 19.32), located near the center of the metropolitan area.

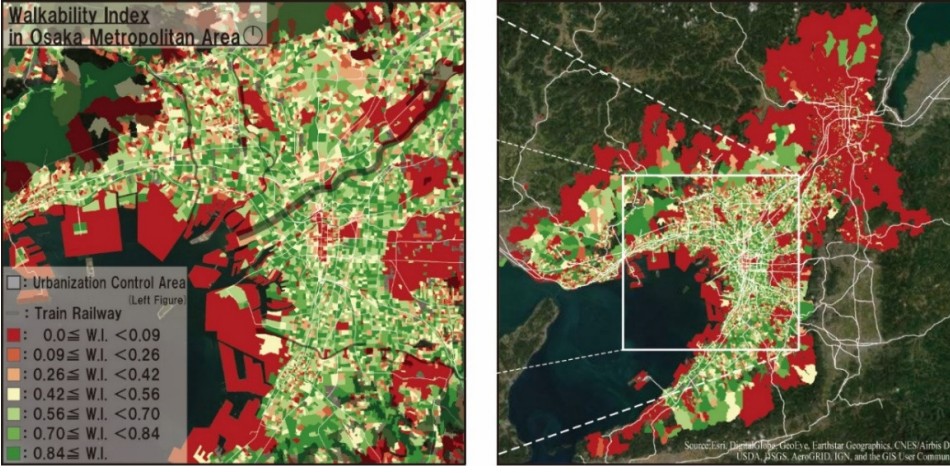

**Figure 3.** Walkability Index (WI) in the Osaka Metropolitan area.

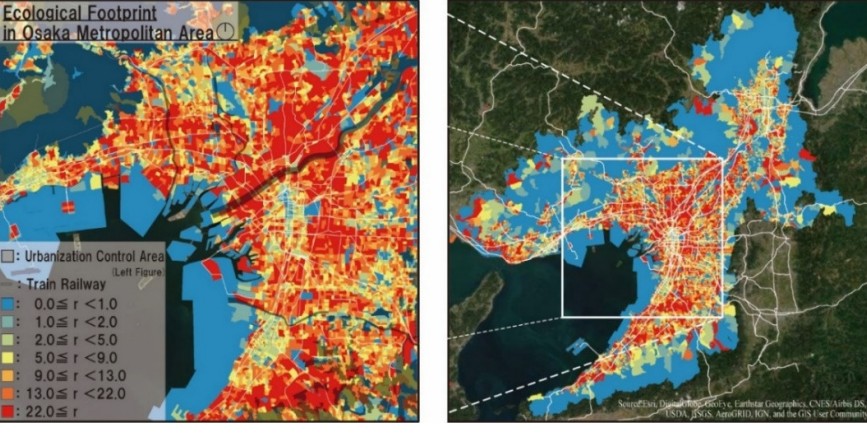

**Figure 4.** Ecological Footprint to Biocapacity (EF/BC) ratio in the Osaka Metropolitan area.

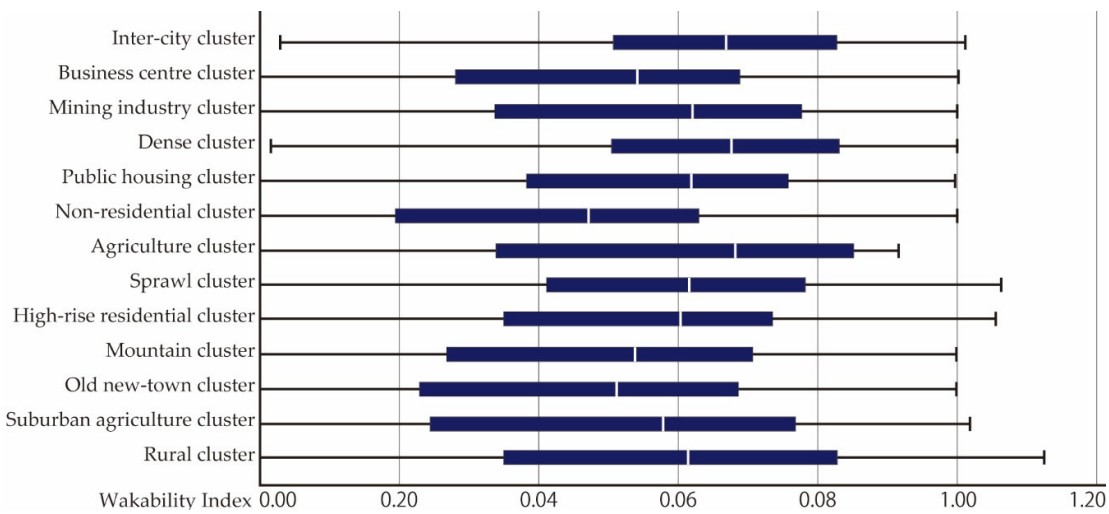

**Figure 5.** WI of each residential cluster.

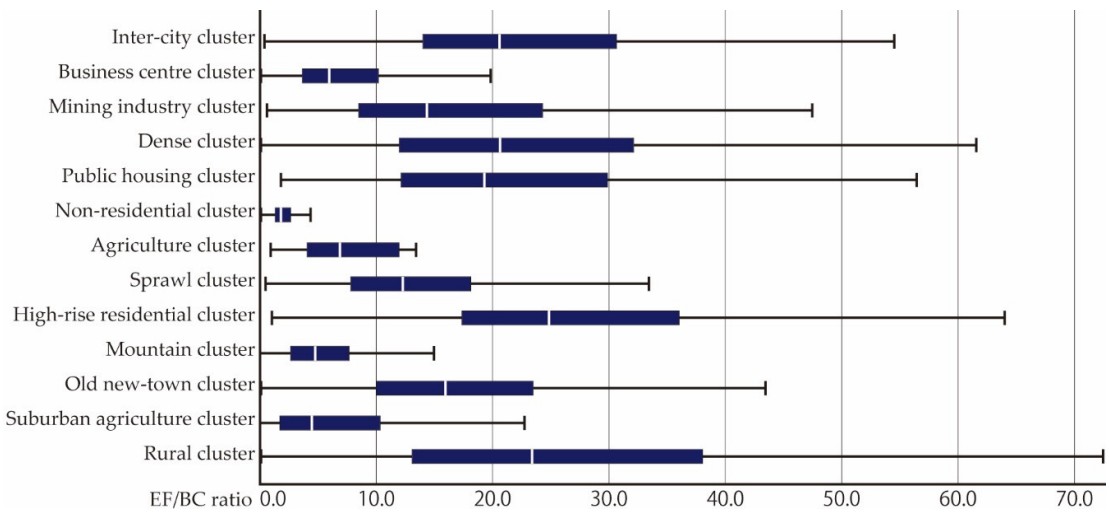

**Figure 6.** EF/BC ratio of each residential cluster.

### 3.3. Statistical Relationship between WI and EF/BC Ratio

The statistical causal relationship between WI and EF/BC ratio was clarified based on the results of regression analyses (stepwise selection method) in Section 2.4 (Table 1). Table 1 shows only the results of the variables adopted in the stepwise selection method. As a result, Table 1 shows that causal relationships differ depending on the type of residential cluster. For example, it was found that WI negatively affects the EF/BC ratio in the sprawl cluster (B = −40.410). This result means that, in the sprawl cluster, increasing WI decreases the amount of excess environmental load. Alternately, Table 1 shows the residential clusters in which WI positively affects the EF/BC ratio. These are inner-city clusters (B = 2.474), non-residential cluster (B = 0.220), old new-town cluster (B = 303.373), and rural cluster (B = 3.591). Among them, the old new-town cluster has a profound influence (B = 303.373). The result means that, in the old new-town cluster, increasing WI increases the amount of excess environmental load. Finally, Table 1 shows the residential clusters in which WI did not affect the EF/BC ratio. These are business center clusters, mining industry clusters, dense clusters, public housing clusters, agriculture clusters, high-rise residential clusters, mountain clusters, and suburban agriculture clusters.

**Table 1.** Statistical Relationship between WI and EF/BC ratio.

| Type of Cluster | Explanatory Variable | Regression Coefficient (B) | Standard Error | *p*-Value |
|---|---|---|---|---|
| Inter-city cluster | WI | 2.474 | 0.864 | 0.004 |
|  | (Constant) | 27.159 | 0.904 | 0.000 |
| Business center cluster | - | - | - | - |
| Mining industry cluster | - | - | - | - |
| Dense cluster | - | - | - | - |
| Public housing cluster | - | - | - | - |
| Non-residential cluster | WI | 0.220 | 0.053 | 0.000 |
|  | (Constant) | 2.544 | 0.062 | 0.000 |
| Agriculture cluster | - | - | - | - |
| Sprawl cluster | WI | -40.410 | 17.510 | 0.021 |
|  | (Constant) | 44.735 | 16.956 | 0.008 |
| High-rise residential cluster | - | - | - | - |
| Mountain cluster | - | - | - | - |
| Old new-town cluster | WI | 303.373 | 34.361 | 0.000 |
|  | (Constant) | 73.952 | 43.848 | 0.092 |
| Suburban agriculture cluster | - | - | - | - |
| Rural cluster | WI | 3.591 | 1.532 | 0.020 |
|  | (Constant) | 31.081 | 2.108 | 0.000 |

"-" are the variables that were not adopted in the stepwise selection method.

## 4. Discussion

The conclusion of this study clarified the effect of walkability on urban sustainability depending on the types of residential clusters in the Osaka Metropolitan fringe area. Specifically, it was found that WI negatively affected the EF/BC ratio in the sprawl cluster. This means that, in the sprawl cluster, improving WI contributes to improving urban sustainability. The negative causality is an interesting result, since both the WI component and the EF component have data related to population, such as household density. The results dictate the points made by Speck [17], who noted qualitatively that improving walkability contributes to urban sustainability. The result also suggests that, in the sprawl cluster, active efforts to improve WI can contribute to the realization of SDGs, which is Goal 11: "Make cities and human settlements inclusive, safe, resilient and sustainable". In other words, the burden on the environment will be reduced by creating walkable living neighborhoods where the elderly can live independently in Japan. Besides, the walkable living neighborhood design is expected to be more collaborative and effective than redevelopment with clearances. Therefore, the design of a walkable living neighborhood is expected to be a way to simultaneously improve the quality of life of its residents and urban sustainability. For example, it is possible to propose methods of attracting commercial land or public facility land to vacant lands that are diffusely found in the sprawl cluster. Coughenour [34] also suggests the need for urban designs that incorporate the characteristics of sprawl, such as access to transit, which has a positive impact on active transport. In addition, improvement of WI is expected to be an effective method in sprawl urban areas where most residents are older adults, because improving WI contributes to maintaining the health of residents.

Additionally, it was found that WI has a positive effect on the EF/BC ratio in the inner-city, non-residential, old new-town, and rural clusters. Among them, in the old new-town cluster, WI has a strong positive effect on the EF/BC ratio. The findings also suggest that, in the old new-town cluster, improving WI may reduce urban sustainability. The result is a different result than what urban planners intuitively felt. In addition, urban planners involved in walkable design needs to face the

inconvenient results. Therefore, the results are novel conclusions elucidated by this study, which was analyzed quantitatively data analysis rather than qualitatively.

In future research, it is crucial to clarify the reasons why different causal relationships occurred depending on the types of residential clusters. The reason is that walkability varies according to the types of residential clusters [35]. Specifically, one possible reason is the relationship between WI and EF/BC ratio. Figures 5 and 6 show that the sprawl cluster has high WI and low EF/BC ratio and the old new-town cluster has low WI and high EF/BC ratio. To elucidate the factors, Appendix C show the mean values of $FP^k_{cropland}$, $FP^k_{grazing}$, $FP^k_{forest}$, $FP^k_{build-up}$, $FP^k_{Carbon}$, and $BC^k$ in each cluster. Appendix C shows that life-related $FP^k_{Carbon}$ was higher in the old new-town cluster than in the sprawl cluster. Among the $FP^k_{Carbon}$, the $FP^k_{ctrans.}$ related to traffic behavior may have influenced the results with the different causal relationship, though detailed investigations are needed in the future, using improved sustainability indicators that take into account micro-scale sustainability as well as macro-scale sustainability. $FP^k_{ctrans.}$ is calculated based on household types and the number of cars used. Therefore, $FP^k_{ctrans.}$ tends to be affected by the social attributes of each residence cluster. For example, sprawl cluster is populated by people of diverse social attributes. On the other hand, the old new-town cluster is populated by high-income workers, who use multiple private vehicles rather than walking. Besides, Kato, et al. [1] suggested that the sprawl cluster is a walkable area. The transportation behaviors of residents in the sprawl cluster are epitomized by walking and cycling. That is because the sprawl clusters are a maze of narrow streets, and it is more convenient for walking and cycling than to drive a car or take public transportation [36]. Kato, et al. [36] clarified that residents walk with confidence because narrow roads have a lower risk of car accidents in the sprawl cluster. Additionally, Kato, et al. [1] indicated that the old new-town cluster is not a walkable area because the old new-town cluster was located on the hillside in the mountains. Therefore, the modes of transportation used by residents in the old new-town cluster primarily include driving by car. In addition, older adults living in the old new-town cluster have difficulty changing from driving cars to walking. However, the residents of the old new-town cluster and the residents of the sprawl cluster, will experience typical health decline associated with aging in Japan [32]. For example, residents are expected to have trouble driving vehicles ultimately. Therefore, the old new-town cluster must realize an independent life by approaches different from the improvement of WI based on the existing concept. For example, some advanced old new-town cluster considers using technologies for Society 5.0, such as automated driving technology [37]. The old new-town cluster is likely to be able to take advantage of these technologies because of their well-equipped infrastructure. Therefore, in the old new-town cluster, the results suggest the likelihood of improving urban sustainability by approaches other than improving WI. In other words, it is a different approach than WI, which is composed of net residential density, density of street connectivity, and land use mix. This is because there may be a different concept of the Walkability, which is "relevant to residential environments that promote walking or cycling with safety, comfort, and the attractions of daily life [1]".

In Japan, the Japanese MLIT actively recommends efforts to improve walkability [8]. That is because the Japanese MLIT has tried to re-design cities based on walkable living neighborhoods where people can live within walking distance to prepare for the inevitable population decline in the future. This study clarified the effectiveness of ongoing efforts to improve walkability in sprawl clusters. In developed countries where sprawl clusters are located over large areas in the metropolitan area, efforts to improve walkability are expected to be an effective method. Especially in developed countries such as China, which are expected to face aging problems next to Japan, improving walkability could be an effective method because it would also contribute to the older adults' health. Zhang [38] clarified built environments to improve the walking activity of the elderly in the Zhongshan Metropolitan area. This result contributes to advancing the understanding of environmental sustainability in order to serve as a bridge between academic and practitioner communities and guide policy and management practice to achieve environmental sustainability. However, some residential clusters, such as the old new-town clusters, have found that the improvement of walkability may negatively affect urban sustainability. For these residential clusters,

this study suggests a need to improve urban sustainability through approaches other than enhancing Walkability. Alternatively, in developing countries where the population will continue to grow, it is better to consider the possibility of not building suburban residential areas in hilly areas. That is because it will be challenging to maintain residential areas like the old new-town cluster in Japan. For the realization of Goal 11 (Make cities and human settlements inclusive, safe, resilient, and sustainable) in SDGs, measures according to the characteristics of residential clusters should be tried. Such attempts will explore plausible alternatives for the future, potentially aiding community planners in constructing living spaces that are designed to improve and maintain the health of residents.

**Funding:** This research was funded by the Obayashi Foundation (Grant number 2019-24) and JSPS KAKENHI (Grant number 19K23558).

**Acknowledgements:** This paper was provided with data from the Conservation GIS-consortium Japan.

**Conflicts of Interest:** The authors declare no conflicts of interest.

## Appendix A

**Table A1.** Content Composition Ratio $R_{ki}$ of Each Indicator for Each Residential Cluster.

| | Inner-City Cluster | Business Center Cluster | Mining Industry Cluster | Dense Cluster | Public Housing Cluster | Non-Residential Cluster | Agriculture Cluster | Sprawl Cluster | High-Rise Residential | Mountain Cluster | Old New-Town Cluster | Suburban Agriculture | Rural Cluster |
|---|---|---|---|---|---|---|---|---|---|---|---|---|---|
| N | 1937 | 5472 | 728 | 672 | 889 | 7403 | 297 | 4998 | 628 | 7251 | 2546 | 2914 | 1033 |
| Urbanized area ratio (%) | 84.5 | 86.2 | 45.1 | 77.1 | 72.4 | 55.2 | 23.6 | 66.2 | 61.8 | 40.7 | 59.1 | 21.3 | 24.9 |
| Average distance from the center (km) | 19.3 | 34.5 | 59.9 | 31.2 | 26.7 | 43.9 | 71.6 | 38.2 | 25.9 | 56.1 | 26.7 | 63.1 | 52.1 |
| Pop. under 15 years old (%) | 0.06 | 0.01 | 0.03 | 0.06 | 0.05 | 0.00 | 0.04 | 0.02 | 0.09 | 0.01 | 0.03 | 0.01 | 0.10 |
| Pop. between 16 and 64 years old (%) | 0.09 | 0.02 | 0.04 | 0.09 | 0.08 | 0.00 | 0.06 | 0.03 | 0.12 | 0.01 | 0.05 | 0.02 | 0.14 |
| Pop. over 65 years old (%) | 0.10 | 0.02 | 0.04 | 0.07 | 0.12 | 0.00 | 0.09 | 0.04 | 0.11 | 0.01 | 0.05 | 0.03 | 0.15 |
| Pop. of Foreigners (%) | 0.09 | 0.02 | 0.02 | 0.04 | 0.07 | 0.00 | 0.01 | 0.02 | 0.03 | 0.00 | 0.01 | 0.00 | 0.05 |
| Pop. who live in their own house (%) | 0.09 | 0.02 | 0.04 | 0.07 | 0.05 | 0.00 | 0.07 | 0.03 | 0.13 | 0.01 | 0.05 | 0.02 | 0.15 |
| Pop. who live in public housing (%) | 0.01 | 0.00 | 0.00 | 0.01 | 0.12 | 0.00 | 0.01 | 0.00 | 0.01 | 0.00 | 0.00 | 0.00 | 0.01 |
| Pop. who live in private rented houses (%) | 0.10 | 0.02 | 0.02 | 0.10 | 0.02 | 0.00 | 0.02 | 0.03 | 0.05 | 0.00 | 0.02 | 0.00 | 0.07 |
| Pop. who live in houses for employees (%) | 0.02 | 0.00 | 0.01 | 0.08 | 0.01 | 0.00 | 0.01 | 0.01 | 0.02 | 0.00 | 0.01 | 0.00 | 0.02 |
| Pop. who live in shared houses (%) | 0.08 | 0.02 | 0.02 | 0.05 | 0.03 | 0.00 | 0.04 | 0.03 | 0.06 | 0.01 | 0.03 | 0.01 | 0.09 |
| HHs who live outside of houses (%) | 0.00 | 0.00 | 0.00 | 0.03 | 0.00 | 0.00 | 0.00 | 0.00 | 0.00 | 0.00 | 0.00 | 0.00 | 0.01 |
| HHs who live in detached houses (%) | 0.07 | 0.02 | 0.05 | 0.05 | 0.03 | 0.00 | 0.10 | 0.04 | 0.09 | 0.02 | 0.06 | 0.03 | 0.19 |
| HHs who live in traditional nagaya-houses (%) | 0.08 | 0.01 | 0.02 | 0.03 | 0.02 | 0.00 | 0.02 | 0.03 | 0.02 | 0.00 | 0.01 | 0.00 | 0.07 |
| HHs who live in apartment houses (%) | 0.08 | 0.02 | 0.02 | 0.08 | 0.10 | 0.00 | 0.01 | 0.02 | 0.08 | 0.00 | 0.02 | 0.00 | 0.04 |
| HHs who live in 1- or 2- stories building (%) | 0.05 | 0.02 | 0.04 | 0.07 | 0.02 | 0.00 | 0.04 | 0.05 | 0.05 | 0.00 | 0.03 | 0.01 | 0.16 |
| HHs who live in 3- to 5- stories building (%) | 0.05 | 0.01 | 0.01 | 0.07 | 0.11 | 0.00 | 0.01 | 0.02 | 0.05 | 0.00 | 0.01 | 0.00 | 0.03 |
| HHs who live in 6- to 10- stories building (%) | 0.06 | 0.01 | 0.01 | 0.06 | 0.04 | 0.00 | 0.00 | 0.01 | 0.05 | 0.00 | 0.01 | 0.00 | 0.02 |
| HHs who live in 11- or more stories building (%) | 0.03 | 0.00 | 0.00 | 0.02 | 0.04 | 0.00 | 0.00 | 0.00 | 0.04 | 0.00 | 0.01 | 0.00 | 0.00 |
| Pop. who work in agriculture and forestry (%) | 0.00 | 0.00 | 0.01 | 0.00 | 0.00 | 0.00 | 0.13 | 0.00 | 0.01 | 0.01 | 0.00 | 0.03 | 0.03 |
| Pop. who work in fishery (%) | 0.00 | 0.00 | 0.01 | 0.00 | 0.00 | 0.00 | 0.04 | 0.00 | 0.00 | 0.00 | 0.00 | 0.00 | 0.00 |
| Pop. who work in mining industry (%) | 0.00 | 0.00 | 0.09 | 0.00 | 0.00 | 0.00 | 0.05 | 0.00 | 0.00 | 0.00 | 0.00 | 0.00 | 0.01 |
| Pop. who work in construction industry (%) | 0.08 | 0.01 | 0.04 | 0.06 | 0.07 | 0.00 | 0.07 | 0.04 | 0.09 | 0.01 | 0.04 | 0.02 | 0.15 |
| Pop. who work in manufacturing industry (%) | 0.05 | 0.01 | 0.03 | 0.07 | 0.05 | 0.00 | 0.05 | 0.03 | 0.08 | 0.01 | 0.03 | 0.02 | 0.12 |
| Pop. who work in electricity, gas, and water supply (%) | 0.01 | 0.00 | 0.01 | 0.02 | 0.01 | 0.00 | 0.01 | 0.01 | 0.03 | 0.00 | 0.01 | 0.00 | 0.03 |

| | | | | | | | | | | | | | |
|---|---|---|---|---|---|---|---|---|---|---|---|---|---|
| Pop. who work in the information industry (%) | 0.08 | 0.01 | 0.02 | 0.08 | 0.04 | 0.00 | 0.01 | 0.02 | 0.12 | 0.00 | 0.04 | 0.00 | 0.06 |
| Pop. who work in transport industry (%) | 0.08 | 0.02 | 0.03 | 0.07 | 0.10 | 0.00 | 0.05 | 0.04 | 0.09 | 0.01 | 0.04 | 0.02 | 0.14 |
| Pop. who work in retail industry (%) | 0.09 | 0.02 | 0.03 | 0.08 | 0.07 | 0.00 | 0.06 | 0.03 | 0.12 | 0.01 | 0.04 | 0.02 | 0.13 |
| Pop. who work in the financial industry (%) | 0.06 | 0.01 | 0.02 | 0.08 | 0.04 | 0.00 | 0.03 | 0.02 | 0.12 | 0.00 | 0.04 | 0.01 | 0.08 |
| Pop. who work in real estate business (%) | 0.09 | 0.02 | 0.03 | 0.08 | 0.06 | 0.00 | 0.02 | 0.03 | 0.12 | 0.01 | 0.04 | 0.01 | 0.08 |
| Pop. who work as researchers or professionals (%) | 0.06 | 0.01 | 0.02 | 0.07 | 0.04 | 0.00 | 0.03 | 0.02 | 0.11 | 0.01 | 0.04 | 0.01 | 0.07 |
| Pop. who work in the service industry (%) | 0.09 | 0.02 | 0.03 | 0.07 | 0.07 | 0.00 | 0.06 | 0.03 | 0.09 | 0.01 | 0.04 | 0.01 | 0.12 |
| Pop. who work in the entertainment industry (%) | 0.06 | 0.01 | 0.02 | 0.05 | 0.05 | 0.00 | 0.04 | 0.02 | 0.07 | 0.01 | 0.03 | 0.01 | 0.09 |
| Pop. who work in education (%) | 0.06 | 0.02 | 0.03 | 0.08 | 0.04 | 0.00 | 0.05 | 0.02 | 0.13 | 0.01 | 0.05 | 0.02 | 0.10 |
| Pop. who work in medical/welfare (%) | 0.07 | 0.02 | 0.03 | 0.07 | 0.07 | 0.00 | 0.06 | 0.03 | 0.11 | 0.01 | 0.04 | 0.02 | 0.13 |
| Pop. who work in joint service industry (%) | 0.05 | 0.01 | 0.05 | 0.06 | 0.05 | 0.00 | 0.18 | 0.03 | 0.09 | 0.02 | 0.04 | 0.05 | 0.18 |
| Pop. who work in other service industry (%) | 0.09 | 0.02 | 0.04 | 0.08 | 0.10 | 0.00 | 0.06 | 0.04 | 0.11 | 0.01 | 0.04 | 0.02 | 0.14 |
| Pop. who work as civil servants (%) | 0.02 | 0.00 | 0.01 | 0.05 | 0.01 | 0.00 | 0.02 | 0.01 | 0.04 | 0.00 | 0.01 | 0.01 | 0.04 |
| Pop. who work at home (%) | 0.08 | 0.02 | 0.04 | 0.06 | 0.04 | 0.00 | 0.18 | 0.03 | 0.07 | 0.01 | 0.03 | 0.05 | 0.13 |
| Pop. who work in their own city (%) | 0.06 | 0.01 | 0.04 | 0.07 | 0.06 | 0.00 | 0.07 | 0.03 | 0.07 | 0.01 | 0.03 | 0.02 | 0.14 |
| Pop. who work in other cities (%) | 0.07 | 0.01 | 0.03 | 0.07 | 0.06 | 0.00 | 0.04 | 0.02 | 0.11 | 0.01 | 0.04 | 0.01 | 0.10 |
| Pop. who work in other wards of their own city (%) | 0.07 | 0.01 | 0.01 | 0.01 | 0.03 | 0.00 | 0.00 | 0.00 | 0.01 | 0.00 | 0.00 | 0.00 | 0.00 |
| Pop. who work in other cities of their own prefecture (%) | 0.03 | 0.00 | 0.02 | 0.07 | 0.05 | 0.00 | 0.05 | 0.03 | 0.09 | 0.01 | 0.04 | 0.01 | 0.12 |
| Pop. who work in other prefectures (%) | 0.03 | 0.01 | 0.02 | 0.07 | 0.03 | 0.00 | 0.01 | 0.01 | 0.15 | 0.00 | 0.05 | 0.01 | 0.05 |
| Pop. who go to sch. in their own city (%) | 0.04 | 0.01 | 0.02 | 0.06 | 0.04 | 0.00 | 0.04 | 0.02 | 0.07 | 0.01 | 0.03 | 0.01 | 0.09 |
| Pop. who go to sch. in other cities (%) | 0.07 | 0.01 | 0.03 | 0.06 | 0.06 | 0.00 | 0.04 | 0.03 | 0.14 | 0.01 | 0.05 | 0.01 | 0.11 |
| Pop. who go to sch. in other wards of their own city (%) | 0.07 | 0.02 | 0.01 | 0.01 | 0.03 | 0.00 | 0.00 | 0.00 | 0.01 | 0.00 | 0.00 | 0.00 | 0.00 |
| Pop. who go to sch. in other cities of own prefecture (%) | 0.04 | 0.01 | 0.03 | 0.06 | 0.06 | 0.00 | 0.06 | 0.03 | 0.13 | 0.01 | 0.05 | 0.02 | 0.13 |
| Pop. who go to sch. in other prefectures (%) | 0.04 | 0.01 | 0.02 | 0.06 | 0.03 | 0.00 | 0.03 | 0.02 | 0.16 | 0.01 | 0.05 | 0.01 | 0.08 |
| Pop. who had lived since birth (%) | 0.07 | 0.02 | 0.05 | 0.06 | 0.05 | 0.00 | 0.13 | 0.04 | 0.10 | 0.02 | 0.04 | 0.05 | 0.17 |
| Pop. who had lived for 1 year (%) | 0.06 | 0.02 | 0.02 | 0.10 | 0.05 | 0.00 | 0.03 | 0.03 | 0.07 | 0.00 | 0.03 | 0.01 | 0.08 |
| Pop. who had lived for the past 5 years (%) | 0.06 | 0.01 | 0.02 | 0.08 | 0.06 | 0.00 | 0.03 | 0.02 | 0.08 | 0.00 | 0.03 | 0.01 | 0.09 |
| Pop. who had lived for the past 10 years (%) | 0.07 | 0.01 | 0.03 | 0.07 | 0.07 | 0.00 | 0.04 | 0.03 | 0.12 | 0.01 | 0.04 | 0.01 | 0.11 |
| Pop. who had lived for the past 20 years (%) | 0.08 | 0.02 | 0.03 | 0.07 | 0.08 | 0.00 | 0.05 | 0.03 | 0.15 | 0.01 | 0.05 | 0.01 | 0.13 |
| Pop. who had lived for over 20 years (%) | 0.07 | 0.02 | 0.04 | 0.05 | 0.08 | 0.00 | 0.08 | 0.03 | 0.09 | 0.01 | 0.05 | 0.02 | 0.15 |

In Appendix A, each number is represented by a green to red graduation. Specifically, the red tab has higher numbers, and the green tab has lower numbers.

## Appendix B

*Appendix B.1. Cropland FP*

$FP_{cropland}^k$ means that cropland is necessary to grow crops for food and feed. The $FP_{cropland}^k$ is composed of $FP_{c.food}^k$, which is cropland required for food/animal feed crop production and $FP_{c.clothing}^k$, which is cropland required for clothing and crop production, in Equations (A1)–(A8). These equations assume that the per capita consumption in the residential area is the same as that in Japan.

$$FP_{cropland}^k = FP_{c.food}^k + FP_{c.clothing}^k \tag{A1}$$

$$FP_{c.food}^k = \sum_{i=11}^{11} \frac{F_{cf.i}^k}{Y_{cf.i}} \tag{A2}$$

$$F_{cf.i}^k = \sum_{P=1}^{10} p_p^k \cdot f_{pi} \tag{A3}$$

$$Y_{cf.i} = \left[ \left\{ \left( \frac{DO_i}{DO_i + VI_i} \right) \times DY_i \right\} + \left\{ \left( \frac{VI_i}{DO_i + VI_i} \right) \times IY_i \right\} \right] \times R_i \tag{A4}$$

$$IY_i = \sum_{N=1}^{a} \frac{VI_{ia} \times IY_{ia}}{VI_{sia}} \tag{A5}$$

$$FP^k_{c.clothing} = FP^k_{c.cotton} + FP^k_{c.wool} \tag{A6}$$

$$FP^k_{c.cotton} = \frac{p^k}{P} \times C_{cc} \times \sum_{a=1}^{5} \frac{W_{cc.a}}{Y_{cc.a}} \tag{A7}$$

$$FP^k_{c.wool} = \frac{p^k}{P} \times \frac{C_w}{K} \times \frac{A}{Y_{cw.b}} \tag{A8}$$

where $F^k_{cf.i}$ represents the consumption of food $i$ in the residential area $k$; $Y_{cf.i}$ is land productivity of food $i$ (t/ha); $p^k_P$ is the population of age group $p$ (0–14 years/15–19 years/20–29 years/30–39 years/40–49 years/50–59 years/60–69 years/70 years or older) in the residential area $k$ by Japanese census data in 2015 [19]; $f_{pi}$ is the consumption of food $i$ in age bracket $p$ by the data of Japanese National Health and Nutrition Survey in 2015 [39] (t); $DO_i$ is the domestic production of food $i$ by the data of Japanese crop statistics survey in 2015 [40] (t); $VI_i$ is the imported amount of food $i$ by the Japanese trade statistics data in 2015 [41] (t); $DY_i$ is the domestic land productivity of food $i$ by the data of Japanese crop statistics survey in 2015 [40] (t/ha); $IY_i$ is imported land productivity of food $i$ (t/ha); $R_i$ is the percentage of the edible portion of food $i$ by the data of Japanese food supply and demand table in 2015 [42] (%); $VI_{ia}$ is import volume of food $i$ from country $a$ by the Japanese trade statistics data in 2015 [41] (t); $IY_{ia}$ is land productivity in the country $a$ of the imported food $i$ by the FAO (The Food and Agriculture Organization) data in 2015 [43] (t/ha); $VI_{sia}$ is the total amount of imported food $i$ up to country $a$ by the Japanese trade statistics data in 2015 [41] (t); $a$ is the ranking of the countries that account for more than 80% of the total imports by the Japanese trade statistics data in 2015 [41]; $FP^k_{c.cotton}$ is cropland required for cotton yarn in residential area $k$ (ha); $FP^k_{c.wool}$ is cropland required for wool in residential area $k$ (ha); $p^k$ is the population in residential area $k$ by Japanese census data in 2015 [19]; $P$ is the population in Japan by Japanese census data in 2015 [19]; $C_{cc}$ is annual domestic consumption of cotton yarn by the statistical data of the Japanese Chemical Fibers Association [44] (t); $Y_{cc.a}$ is land productivity of cotton yarn at import destination $a$ by FAS [45] (t/ha); $W_{cc.a}$ is import ratio at destination $m$ by the Japanese trade statistics data in 2015 [41] (%); $C_w$ is annual domestic consumption of wool by the statistics data of Japanese Chemical Fibers Association [44] (t); $K$ is the amount of wool that can be collected from one sheep by the Japanese Livestock Industry Association data [46] (t); $A$ is the amount of barley required to raise one sheep by the Japanese Livestock Industry Association data [46] (t); and $Y_{cw.b}$ is land productivity of barley by the FAO data in 2015 [43] (t/ha).

*Appendix B.2. Grazing Land FP*

$FP^k_{grazing}$ means that grazing land is necessary to graze animals for meat and milk. $FP^k_{grazing}$ is composed of $FP^k_{g.food}$, which is required for meat and milk production, and $FP^k_{g.wool}$, which is required for wool production. $FP^k_{g.food}$ is calculated using the same method as $FP^k_{c.food}$ in Equation (A9). However, land productivity $Y_i$ is calculated by converting it into the amount of grass. In addition, $FP^k_{g.wool}$ was calculated based on data from the Japanese Livestock Industry Association [46], assuming that a grassland area of 0.5 ha per 10 mature sheep is required for one year in Equation (A10).

$$FP^k_{grazing} = FP^k_{g.food} + FP^k_{g.wool} \tag{A9}$$

$$FP^k_{g.wool} = \frac{p^k}{P} \times \frac{C_w}{K} \times 0.05 \tag{A10}$$

*Appendix B.3. Forest Land FP*

$FP_{forest}^k$ represents the forestland needed to obtain materials for use in paper production in the residential area *k*. $FP_{forest}^k$ is calculated in Equation (A11).

$$FP_{forest}^k = \frac{p^k}{P} \times r_p \times \sum_{l=1}^{7} \frac{W_l}{\beta_l} \tag{A11}$$

where $r_p$ is the household consumption rate of paper (%); $W_l$ is wood pulp chip demanded import destination *l*, which is calculated based on data from the White Paper on Forests and Forestry in Japan [47] (m³); $\beta_l$ is the growing stock amount of forest of destination for import *l*, which was calculated based on the FAO Global Forest Resources Assessment 2015 data [48] (m³/ha). *l* is the ranking of countries that account for more than 80% of total imports in Japan by the Japanese trade statistics data in 2015 [41].

*Appendix B.4. Build-Up Land FP*

$FP_{build-up}^k$ means that build-up land is needed to conduct urban activity in the residential area *k*. $FP_{build-up}^k$ is calculated in Equation (A12).

$$FP_{build-up}^k = \sum_{s=1}^{8} b_s^k \tag{A12}$$

where $b_s^k$ is the area build-up of land use *s* in the residential area *k*, which is calculated from the data of the numerical map 5000 in Japan [28]; *s* is eight types of land uses (industrial land, residential land, commercial land, office land, road land, park, green space, and public facility land).

*Appendix B.5. Carbon FP*

$FP_{Carbon}^k$ means that forestland needs to absorb $CO_2$ from fossil fuels for household and private transport use in residential area *k*. $FP_{Carbon}^k$ consists of $FP_{clife}^k$, which is forest land needed to absorb the $CO_2$ emitted by the household sector in the residential area *k*, and $FP_{ctrans.}^k$, which is forest land needed to absorb the $CO_2$ emitted by the transportation sector in the residential area *k*. $FP_{clife}^k$ is calculated in the equation, and $FP_{ctrans.}^k$ is calculated using Equations (A13)–(A15).

$$FP_{Carbon}^k = FP_{clife}^k + FP_{ctrans.}^k \tag{A13}$$

$$FP_{clife}^k = \sum_{l=1}^{3} \sum_{m=1}^{2} \frac{C_{ml} \times p_{ml}^k}{r_f} \tag{A14}$$

$$FP_{ctrans.}^k = \sum_{x}^{2} \frac{\left( \sum_{x=1}^{2} \sum_{y=1}^{4} C_{xy} \times p_{xy} \right) \times p_y^k}{r_f} \tag{A15}$$

where $C_{ml}$ is $CO_2$ emissions by households of house type *m* in the residential area *k* of city rank *l*, which was calculated from the data of the Statistical Survey of $CO_2$ Emissions in the Home Sector in 2017 [49] (t-$CO_2$); $p_{ml}^k$ is the number of households of house type *m* in the residential area *k* of the city rank *l* by Japanese census data in 2015 [19]. $r_f$ is the $CO_2$ absorption efficiency of forest land (t-$CO_2$/ha); *l* is three types of city rank: (1) cities with prefectural offices and ordinance-designated cities, (2) cities with a population of 50,000 or more, and (3) municipalities with a population of less than 50,000; $C_{xy}$ is $CO_2$ emissions by gasoline household type *x* and number of vehicles used *y*, which is calculated by the data of the Statistical Survey of $CO_2$ Emissions in the Home Sector in 2017 [49] (t-$CO_2$); $p_{xy}^k$ is the ratio of the number of households to the number of vehicles used *y* in household type *x* in the residential area *k* ; $p_y^k$ is the number of households of household type *x* in the residential area *k* by Japanese census data in 2015 [19]; *x* is household types (single household and households of 2 or more people); and *y* is the number of cars used (0, 1, 2, 3, or more).

## Appendix C

**Table A2.** Mean Values of $EF^k$ and $BC^k$ in Each Eluster.

| | Inner-City Cluster | Business Center Cluster | Mining Industry Cluster | Dense Cluster | Public Housing Cluster | Non-Residential Cluster | Agriculture Cluster | Sprawl Cluster | High-Rise Residential Cluster | Mountain Cluster | Old New-Town Cluster | Suburban Agriculture Cluster | Rural Cluster |
|---|---|---|---|---|---|---|---|---|---|---|---|---|---|
| $FP^k_{cropland}$(ha) | 90.1 | 21.1 | 55.7 | 84.6 | 90.4 | 4.7 | 55.2 | 39.9 | 108.1 | 13.2 | 52.0 | 25.7 | 101.1 |
| $FP^k_{grazing}$(ha) | 4.7 | 1.1 | 2.9 | 4.5 | 4.6 | 0.2 | 2.8 | 2.1 | 5.8 | 0.7 | 2.7 | 1.3 | 5.4 |
| $FP^k_{forest}$(ha) | 0.1 | 0.0 | 0.1 | 0.1 | 0.1 | 0.0 | 0.1 | 0.0 | 0.1 | 0.0 | 0.1 | 0.0 | 0.1 |
| $FP^k_{build-up}$(ha) | 250.0 | 211.6 | 214.1 | 223.7 | 225.8 | 180.7 | 236.7 | 199.5 | 213.7 | 160.1 | 186.2 | 164.0 | 221.7 |
| $FP^k_{Carbon}$(ha) | 5101.4 | 1247.6 | 3165.8 | 4618.5 | 4638.2 | 269.1 | 2752.7 | 2289.3 | 5363.1 | 752.3 | 2912.6 | 1317.1 | 5601.8 |
| $BC^k$(ha) | 256.4 | 233.0 | 254.8 | 239.0 | 230.9 | 211.3 | 630.2 | 206.7 | 230.1 | 235.3 | 218.8 | 480.8 | 275.9 |

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
