# Peer review of "Effect of Walkability on Urban Sustainability in the Osaka Metropolitan Fringe Area"

_sustainability, doi:10.3390/su12219248_

Round 1

Reviewer 1 Report

1. What do you mean by "fringe area"? I took this to assume you are only looking at areas outside the central city, but from reading further, I am assuming that you analyzed the entire metro area... But this isn't clear until the results section where you have maps finally say that you only analyze residential clusters that are 50% urbanized and over 10km from the city center. But then your maps never actually identify these areas and instead show the results for the entire region... which then begs the question: if you have data for the entire metropolitan area, why not just write and article that talks about the entire metro area??? Why the focus on "fringe" areas? This is not clear and needs to be resolved.

2. Major methodological assumption that raises a huge red flag for me is that you are applying national level measures of the ecological footprint to neighborhoods - simply using the national averages and applying it to the local area. This however does not take into account that those in lower density suburban areas are likely to be more affluent than the general population, therefore having a larger ecological footprint. This whole study's focus on fringe areas (which I interpret as being more suburban) is a problem. If you do not take into account local variation in the EF, then this study is meaningless. There is a reason you don't see many studies using the EF at such a small scale - because it really isn't useful unless you find ways to measure aspects of it that vary between neighborhoods - like with transportation mode, or adjusting for neighborhood income levels... but none of this was done in this study. The only one of the EF variables I see accounting for local variation is transport, which looks at gasoline consumption x number of cars per household, and does not appear to take into account how far these households drive, which would be the key factor in figuring out how much CO2 they produce. A household could have 10 cars, but drive 1 day a week to the grocery store and they would have the same measure as a household that has 10 cars that each drive 100,000 miles.

3. Additionally, using a ratio of the ecological footprint of a neighborhood compared to its bio-capacity is just counter-intuitive. I don't know many folks that would agree with the statement: all neighborhoods should have a biocapacity equal to their ecological footprint. The very nature of urban areas acknowledges that the biocapacity upon which they rely is not found in the immediate area... So why you would calculate this ratio, I have no idea.

4. While the detailed discussion of the methodology and the description of what every single variable in the ecological footprint is perhaps interesting for some, it is mostly tedious and unnecessary. It could be provided in a table that takes up half a page instead of 3.5 pages of text or as a page in the appendix where it would be a optional read for those who are interested.

5. You spend so much of the methods talking about the EF, that you never talk about the urbanization elements of the areas you are trying to study... how dense are they? i.e. how many people per square km? This seems like it would be an important element when talking about walkability...

6. You comment on page 7: "dense cluster was constructed with too narrow streets with fragile road networks, as well as insufficient public open spaces and high population density." You write this in a way that implies this is a bad urban form, but this is exactly what most walkability scholars would identify as highly walkable... Also, sprawl cluster area being those that lack urban infrastructure, again seems like your subjective opinion because you don't qualify it. In other words, what infrastructure is it missing? why are vacant lots "useless"? For the high-rise cluster, it could be useful to show on a map the railroad lines they are on. These sound like TOD type developments if they are built around commuter rail, which again, represents a walkable urban form.

7. I would suggest in identifying the urban clusters, you highlight each one on a map separately with its image. This also seems more relevant to include in the methods, not the results.

8. Generally high walkability is shown in green and low scores in red. You may want to adhere to the same scale so that someone looking at it doesn't mistake high walkability areas for low ones.

9. Starting on line 294 where you finally get to the "results" you seem to be arguing that people need to alter their lifestyles because of high EF to BC ratios... but this is exactly what you would EXPECT to find. Larger areas that are less dense are going to be better off, just as your data shows. they have a larger land area to absorb human activity. But denser urban areas do not, because there are more people in less area.. meaning that the ratio you calculate will be inevitably higher... but we don't want everyone to live a rural, suburban, or agrarian lifestyle? WHy? Without a more detailed analysis of the variation of EF between these different urban clusters, the data presented here is meaningless. Using the EF/BC ratio is complete non-sense. And no sane researcher would propose using such a metric. Cities are by definition dense places that offload their EF on rural areas. No one says this is entirely a bad thing and plenty of evidence of urban typologies shows that people in these more dense area have lower ecological footprints, but your work here hadn't teased that out...

10. And not to beleaguer the point, but your results on WI are exactly what I would expect as well... Places that are more dense with better road networks are more walkable, and also have less capacity to absorb their own ecological footprints. But are those footprints higher or lower than those in other types of neighborhoods? That is a key part missing in this study. Simply applying averages from the National EF surveys does nothing to understand EF at the neighborhood scale. The findings of the WI do nothing to tell me about this relationship other than to state the absolute obvious... that dense urban neighborhoods where people walk have less capacity to absorb the activities of people living within that neighborhoods borders. This is the very nature of cities and noone says this is bad. What is bad about the EF is the global impact and the fact that increasingly the global ecosystem is unable to absorb the entirety of human activity.

11. Your discussion, based on my reading of your results (which could be wrong since they are very unclear) makes little sense. If your analysis found that denser clusters have a worse EF/BC ratio, how would making a suburban area more walkable (i.e. interpreted by most as more dense) result is a lower EF/BC ratio?? This is entirely counter-intuitive and you offer no context or explanation of why this would be the case. And your suggestion that improving walkability in certain clusters being a negative impact on urban sustainability is completely counter-productive. We want cities to be more walkable and the most walkable places that exist can always be improved upon... and walking positively impacts sustainability outcomes, so if a place becomes more walkable, then the impacts are better off. Your read of the your findings stem from the misguided use of the EF/BC ratio which is inevitably going to be lower in more dense places, but using this micro-scale measure as a proxy for urban sustainability is no good and ignores the fact that sustainability cannot exist in isolated micro-environments, but is a global problem.

12. A note on other ways to address mobility... promoting and increasing walkability doesn't preclude other approaches to improving mobility (i.e. with AVs) or using technology to minimize trips (i.e. remote health care). These approaches should be used, but also have negative impacts on urban sustainability. Deploying more cars would likely encourage more people beyond those who need them to use them and not walk, thus negatively impacting sustainability.

Author Response

Dear Reviewer:

I appreciate the reviewer for the generous comment on the manuscript. I have edited the manuscript according to the comments from editors and reviewers.

I believe that the manuscript is now suitable for publication in Sustainability and look forward to hearing from you concerning your editorial decision.

Yours sincerely

Haruka Kato

Reviewer 2 Report

I consider that the manuscript addresses a very interesting topic for the urban environment. My greetings to the author. However, I think it is necessary to make some amendments so that the paper can be published in "sustainability".

Here are my detailed comments:

In section 2: adding a diagram/figure showing the four steps that were followed would add value to the document.

I think that merging the results section with the discussion section would give greater value to the work. Furthermore, some aspects are already discussed in the results section. Also, I recommend that the author add a final section of conclusions in which only the effects of walkability are defined, without the need to provide further arguments (which should already be pointed out in the previous sections and argued with other references).

On the other hand, in the paper, the SDGs are mentioned on different occasions. However, specifying what specific objectives it refers to would give a more scientific tone and a higher value to the manuscript.

Finally, I see it important to point out that the discussion section is generally the most valued section in a document that aims to show novel knowledge, so I recommend arguing in a better way each of the indicated findings, supporting or debating with references from other works already published.

Author Response

(The authors gave the same response as above.)

Reviewer 3 Report

  1. In my opinion data from 2015 should be updated as much as it is possible
  2. I recommend to use passive voice :
    1. line 102: First, this section analyzed the standardization-> in this section the stabdardization(...) was analyzed
    2. the same in line 245 and 294 (Section X.X analyzes )
  3. In my opinion, it would be worth to compare Osaka with others “walkable city”, it will make the article more universal
  4. The article is very interesting, but it should be more universal
    message and be focused more on comparing cities,
    less on equations, especially the author use 4 auto citation with a similar problem.

Author Response

(The authors gave the same response as above.)

Round 2

Reviewer 1 Report

Surprisingly, the few changes you made do improve the paper. But there are still many shortcomings. The maps are all much more clear in terms of where is walkable and where is less walkable. There are several points in the discussion that I feel you could still elaborate on, especially where you mention transportation options available to people in different neighborhood types. Line 287 seems to have an incomplete thought where you reference Speck "who noted qualitatively" This thought could be expanded as I do not have a clear understanding of what you mean by it the way it is currently worded.

Likewise, in lines 297-8 you mention your findings again being quantitative instead of qualitative... but I could also say you need a stronger discussion of why your quantitative methods and analysis may be off. Are there other quantitative components of walkability or EF or BC that could have been measured that may have altered your findings? You get into this a little bit around line 316 where you start to talk about transportation mode and walkability. But the very brief inserted sentences you made barely begin to address some of these important issues. I think you could also expand on the discussion of the SDG number 11. You really just add a sentence that mentions it without actually discussing it.

My big issue though is that you responded in great detail to my comments but then added so little to the paper itself. A sentence here or there does not begin to address some of the major problems I see with this paper. In one of your comments, you mention how your results are counter-intuitive but that we should not shy away with inconvenient truths. I entirely agree with this. BUT you have not convinced me in this paper that these counter-intuitive results are not a result of the way you are measuring these variables. In other words, I am not convinced by your methodological approach to measuring the relationships between walkability and sustainability (as measured by EF/BC ratio). If you included a more thorough assessment of the shortcomings of your methodology and what might need to be done differently in subsequent work to validate or disprove your findings presented here. I do not think what you have added does enough to address these concerns. I think another way you could address this is to add more justifying why your EF/BC ratio is a good and valuable proxy for urban sustainability. I think this is where we fundamentally disagree because I do not think this is a good proxy and what you have shown in the paper does not convince the reader that it is. As such, I am more skeptical of your results than I might be otherwise.

One final comment, it would be nice to include at least some discussion of how this research is useful to other places. You haven't done much to discuss the specific context of Japan and you mention this work is relevant to cities in other developed countries, but never specify how or why this is the case. Since this is an international journal, this may be good to expand on just a little bit - particularly since it is focused on urban sprawl areas of the metropolitan area. There is no shortage of those in the U.S. and there's lots to reference in regard to making suburbia more walkable - a vast literature that you never touch in your discussion.

Author Response

(The authors gave the same response as above.)
